# Eco-Friendly Processing of Wool and Sustainable Valorization of This Natural Bioresource

**Crisan Popescu [1] and Michaela Dina Stanescu [2,\*]**

[1] KAO Germany GmbH, Pfungstaedter Str. 98-100, 64297 Darmstadt, Germany; crisan717@yahoo.co.uk

[2] Department of Natural and Technical Sciences, Aurel Vlaicu University, 77 Revolutiei Blvd., 310130 Arad, Romania

\* Correspondence: stanescu@uav.ro

**Abstract:** The environmental invasion of plastic waste leads to, among other things, a reassessment of natural fibers. Environmental pollution has shown the importance of the degradability, among other properties, of the raw materials used by the textile industry or other industrial fields. Wool seems to be a better raw material than the polymers that generate large quantities of micro- and nano-plastics, polluting the soil, water, and air. However, the usual processing of raw wool involves a number of chemically very polluting treatments. Thus, sustainable procedures for making wool processing environmentally friendly have been considered, leading to the reappraisal of wool as a suitable raw material. Besides their applications for textile products (including *smart textiles*), new directions for the valorization of this natural material have been developed. According to the recent literature, wool may be successfully used as a thermal and phonic insulator, fertilizer, or component for industrial devices, or in medical applications, etc. In addition, the wool protein α-keratin may be extracted and used for new biomaterials with many practical applications in various fields. This review makes a survey of the recent data in the literature concerning wool production, processing, and applications, emphasizing the environmental aspects and pointing to solutions generating sustainable development.

**Keywords:** sustainable development in sheep breeding; wool a natural raw material; sustainable enzymatic wool processing; wool sustainable valorization

## 1. Introduction

The pollution generated by micro- and nano-plastics, spread all around us, and disturbing the flora and fauna, has changed attitudes concerning the utilization of plastic materials [1–4]. Thus, in the textile field, interest has turned back to the natural fibers of vegetal (bast, cotton, etc.) and animal (silk and wool) origin [1].

Wool has accompanied mankind since the dawn of the latter. It features in legends ("golden fleece"), the Bible, and historical facts (The Woolsack in the Parliament of the UK). Over the centuries, wool has been an important fiber for textile producers. The products have ranged in quality and price from the relatively inexpensive to the ultra-luxurious scarlet woolens aimed at aristocratic markets [5].

Wool is sheep hair, and it is a raw material with many qualities, like elasticity, moisture management, good heat insulation due to air retention, gentle luster, etc. [6]. Wool products are comfortable to wear and are considered by consumers to be a renewable and environmentally friendly material. The study of Collie and coworkers [7] evidenced the high level of wool-product biodegradation in seawater (more than 20%), compared with cotton (10%) and synthetic fibers (less than 1%) during the same period. Wearing wool brings many benefits such as reduced body odor, improved sleep quality, protection against fire, and reduced eczema symptoms [8–10]. Despite all these excellent

properties, wool is today only about 1% of the total amount of consumed fibers [11], as illustrated in Figure 1.

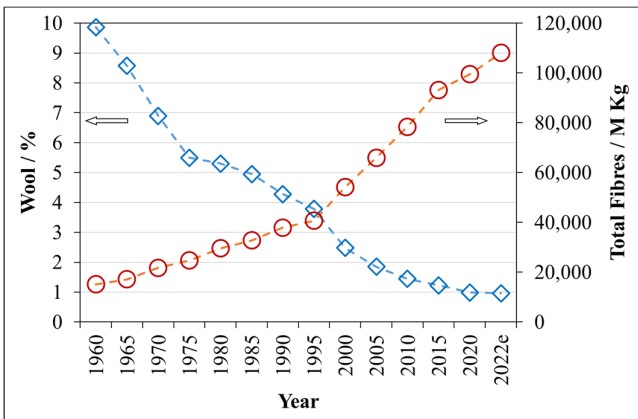

**Figure 1.** World fiber consumption evolution (blue-wool; red-total fibres) [11] (2022e: estimated value for 2022).

This situation is created by the complexity of wool processing, generating high water and carbon footprints [12]. The implementation of a circular economy policy requires serious investments, the pay-off taking time [13]. Indicators for comparing the Life Cycle Assessments (LCAs) of natural and synthetic fibers [14] and the products from such fibers [15,16] have been appraised.

The environmental impact depends also on the type of wool processed [17]. New indicators have been introduced for measuring wool, like *ZQ*, involving animal welfare, environmental sustainability, fiber quality, and the traceability of products, as well as social responsibility, together with *ZQRX* (regenerative index), measuring how well the producers compensate carbon, restore watercourses, and protect native species [18]. The *eco-label*, established in 2014 by the EU Commission [19], is another criterion for comparing the environmental impacts of different textile products [20]. Such labels stimulate producers and consumers to produce and, respectively, to buy *ecofriendly* products [21].

These environmental problems have appeared alongside the evolution of sheep farming due to food (land) and water consumption and pollution harms. The greenhouse effect is largely due to methane releases [22]. Wiedemann and coworkers [23] investigated the environmental effects of farming for wool supply and the production of woolen sweaters. Energy consumption is determinant as to wool processing and greenhouse emissions at the farming stage. But those environmental costs have to be considered alongside the benefits of other sheep products like milk and cheese, as well as meat. [24,25].

The economic analysis revealed that sheep breeding is not very efficient, but the social and cultural aspects, the traditions, the community cohesion, also have to be put in balance [26]. Further on, the trail of wool along the value chain, from farms to spinning mills and finished product, is important [27].

As is generally known, clothes are, together with food and shelter, basic needs for everyone. The fabrication of textiles for clothing involves the production and processing of fibers, followed by the finishing operations. All of these steps require water and chemicals consumption, generating pollution [28]. Natural fibers, such as wool, consume, for their processing, more water than do synthetic fibers. The water consumption for wool fabric is 200–300 kg/kg, less than for cotton (250–350 kg/kg), but more than the amount used for synthetic polyester (100–200 kg/kg) [29].

The evaluation of wool's value also has to consider that, in addition to its uses in textile material manufacture, it may be valorized in construction, agriculture, cosmetics, etc.

The processing of wool fibers into textiles plays an important role in the pollution of the environment, and several solutions have been applied in efforts to reduce this. This paper aims mainly to survey the enzymatic processes available for wool preparation and finishing as a clean alternative to the traditional procedures, as well as some non-traditional and sustainable procedures for the valorization of wool.

## 2. Sustainable Development in Sheep Breeding

Farming systems may be cost-effective, diminishing the pressure to choose between biodiversity preservation and economic development [30]. Modern technology may also help animal husbandry farms to become cost-effective [31–33]. The level of technology adoption determines the increases in the ecological aspects of the pastoral system [34].

Several papers provide information regarding the efficient management of sheep breeding. There are many factors influencing its efficiency, like the relations with stakeholders [35], socio-technical networks in the field [27], animal genetics [36], animal welfare [33], flock management [37], and pasture productivity [38].

The implementation of a correct EU Common Agriculture Policy for supporting the pastoral economy is also of importance. Determining the correct value chain of the products and having a right appreciation of the profession, which can attract young people by introducing sustainable financial facilities, are compulsory [39]. Integration of crops and sheep breeding is of importance for land-use efficiency, improvements in productivity, and decrease in the environmental burden [40]. Also, special computer programs like Smart Sheep Breeder applied in India [41] have been developed for improving decision making in sheep breeding.

## 3. Wool Structure and Properties

Wool, like other animal hairs, is a natural composite system, one which has a complex dual structure at all levels. The whole fiber appears like a ring/core arrangement, with the cortex wrapped by the cuticle. The cortex contains ortho- and para-cortical cells and the cell membrane complex. The cortical cells are further composed of macrofibrils and intermacrofibrillar material. The macrofibrils consist of keratin intermediate filaments (KIF) and an intermicrofibrillar matrix made of keratin-associated proteins (KAP) and cytoplasmatic, as well as nuclear, remnants. As such, the cortex matches, at all levels, the 'filament in matrix' model. By considering the lateral interactions between the KIFs and the surrounding matrix as an interphase, the more complex view of wool uses the three-phase model, which provides better explanations of fiber mechanics and reactions with chemicals [42–44].

Chemically, wool is a protein fiber made of $\alpha$-keratin, a fibrous protein. The elemental analysis of wool shows carbon, hydrogen, oxygen, nitrogen, and sulfur. The high sulfur percentage (around 5 wt%) results from the cystine content of the fiber. Total hydrolysis of the peptide bonds in proteins yields the 20 common natural $\alpha$-amino acids found in wool and their mol percentages, which are approximated by the values indicated in parentheses, namely, alanine (5%), arginine (7.2%), asparagine and aspartic acid (6%), cystine (11.2%), glutamine and glutamic acid (12.1%), glycine (8.1%), histidine (0.7%), iso-leucine (2.8%), leucine (6.9%), lysine (2.3%), methionine (0.5%), phenylalanine (2.5%), proline (7.5%), serine (10.2%), threonine (6.5%), tryptophan (1.2%), tyrosine (4.2%), and valine (5.1%) [45].

The peptide arrangement in wool fiber has been a topic of study since the early use of X-rays for investigating protein structures [46,47]. The X-ray diffraction pattern of wool shows a meridian reflection at 0.51 nm and an equatorial reflection at 0.98 nm. Interpreting these results, the $\alpha$-helical structure was proposed to give an account of the secondary structure of the keratin fiber [48]. The organization of the $\alpha$-helices in keratin intermediate filaments, KIF, which form the macrofibrils, the ordered (crystalline) part of the wool fiber, is driven entropically and follows a lateral and end-to-end association (polymerization) rule, as illustrated in Scheme 1 [49].

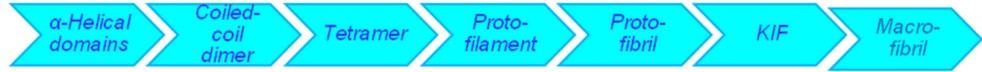

**Scheme 1.** Keratin $\alpha$-helices organized to form macrofibrils [49].

Wool fibers, like any other animal fiber, have a slightly elliptic cross-section and are protected by the scales of cuticle arranged on their surface like tiles on a roof. These scales cause wool to have different fiber–fiber friction coefficients when measured along or against the scale orientation. This property of a differential friction coefficient is unique among textile fibers and is the reason for the felting ability of wool fibers [45].

Wool fibers also contain lipids. The wool grease is a complex mixture produced by sebaceous glands in sheep skin. In addition to this external grease, there are internal lipids trapped in the mass of wool proteins [50].

## 4. Wool Availability

Keratin fibers are available almost everywhere in the world, and wool produced by sheep is by far the most widely used keratin fiber. The total amount of wool produced yearly fluctuates with market demands but decreased over the last 70 years to around 1100 M Kg [11], as indicated by Figure 2.

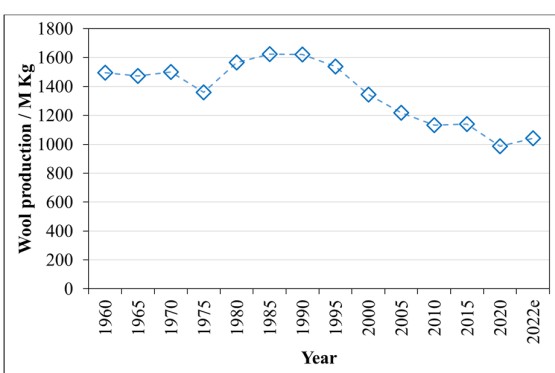

**Figure 2.** Wool, yearly production [11] (2022e: estimated value for 2022).

Economically the sheep is a green factory without wastes: consuming only water and grass as raw materials, a sheep produces wool, milk, and meat, and even the excrement is useful for fertilizing pastures. Because the sheep graze, not pulling the grass as the goats do, the pasture regenerates quickly, within weeks after the passage of sheep. Very roughly, a sheep produces annually 1 kg greasy wool (which means some 0.6 kg clean fibers) from 1 ha of average pasture [51].

## 5. Classical Processing of Wool Fibers

Bringing wool to the textile circuit is a labor-intensive process. The first steps of shearing and collecting the greasy wool, followed by skirting (selecting the parts of the fleece) and classing wool according to the fiber diameter and strength, are manual operations. The collected greasy (raw) wool contains various amounts of impurities collected by the wool during the sheep's grazing, namely sand, dirt, and vegetable matters, to which one adds the grease and suint produced by sheep during its daily biological activity. Depending on the type of wool and the pasture quality, clean wool fibers range from 30 to 75% of the total greasy fleece collected from the sheep.

The numerous chemical and mechanical processes through which the wool fibers are turned into an end-product can be grouped into the following operations:

- Operations of cleaning the raw wool;

- Operations which transform the fibers into fabric;
- Operations of cleaning yarns and fabrics;
- Operations of stabilizing the dimensions of fabrics;
- Operations of surface treatment and infiltration of fibers and fabrics;
- Operations for imparting a finished aspect to fabrics.

The details of the operations differ as to the ways the fabrics were obtained: by weaving or by knitting, respectively. In all of these cases, various chemical products are used for assisting in, or for producing, the required effects. Most operations make extensive use of surfactants for assisting processes, namely, for washing/scouring, for improving the quality of bleaching, of dyeing, or for finishing (i.e., softening and stiffening) the materials. This results in the further production of pollutants, which are either associated with effluents, or emitted into the atmosphere.

### 5.1. Operations of Cleaning the Raw Wool

For the delivery of clean fibers to the industry, the greasy wool goes through scouring, which is the operation of washing the raw fibers with about 1%wt surfactant in a continuous 5–6 bowl line in order to remove the grease, suint, sand, and dirt from the fibers. The waters discharged from the first three bowls used to wash fibers with more than 10% grease (usually from fine wool, Merino type) are further used for separating lanolin, an important ingredient in many personal and health care creams, from the effluent, by means of an Alfa-Laval process [52]. Despite lanolin separation, the wastewater from wool scouring is heavily polluted with organic matters, and much effort is dedicated to processes to clean and reuse them, aiming to reduce water consumption. The recent development of bio-degradable surfactants helps in the initiative to build a more environmentally friendly wool scouring process. Some 5–7 L of water are required to obtain 1 kg of clean wool under severely controlled parameters, but 20 L is more common.

The vegetable matters, if more than 2–3%, have also to be removed before the material is subjected to further processing. This operation, known as "carbonization", makes use of the good resistance of wool to strong acids (particularly sulfuric acid) and of the hydrolysis of cellulose by the same environment. The fibers with vegetable matters (e.g., grass and burrs) are soaked in a 20% sulfuric acid solution, dried and baked, and then crushed to separate the carbonized cellulosic matters from the rest. The fibers are then neutralized and dried. The process produces acid-polluted waters and carbon dust in the air, besides weakening the wool fibers, for which reason alternatives, like the use of enzymes, are sought.

### 5.2. Operations which Transform Fibers into Fabric

This group gathers several mechanical operations which help in the transforming of the fibers into a yarn, namely, carding, combing, and spinning, and further process a yarn into a knitted or woven fabric by knitting or weaving, respectively.

The making of the yarn requires the grouping of fibers into a roving of fairly even density, which is eventually spun. The fibers are kept together in the yarn by fiber–fiber friction, and the twisting movement during spinning ensures a compact packing. Providing that the fibers in the yarn are parallel or randomly arranged by the previous mechanical operations, the yarn is named worsted—its formation consists of a carding operation used to produce a web of fibers, one which is further combed to arrange the fibers in parallel before spinning them into a fine and smooth yarn—or woolen-carded yarn, for the latter of which the combing operation is not used, and the yarn is, thus, bulky. Any of these yarns can then further be used for producing a 3D structure (fabric) by knitting or weaving.

Alternatively, the non-woven fabrics are produced by directly transforming fibers from the carding web into a fabric by felting, without the formation of a yarn. The oper-

ation makes use of the fiber-to-fiber friction of wool in forming a felt. Another way of making an end-product is by punching tufts of wool fibers into a woven support and using resins to bond the ensemble to produce a tufted product, usually a carpet. This felting process is also used for obtaining handcraft objects by using needle felting and lasers for cutting [53,54].

All of these operations use oils (1…2 wt%) for easing the mechanical action and reducing the breakage of the fibers during the process.

### 5.3. Cleaning Operations of Yarns and Fabrics

The *washing/scouring* and the bleaching are the operations used for cleaning the yarns and the fabrics. Scouring makes use of surfactants and warm water (around 40 °C) for washing out the oils and other chemical auxiliaries used in assisting the previous mechanical operations, namely, the spinning, knitting, and weaving. The *bleaching* process is carried out as a last step after washing the materials, and the chemical agents used for it depend on the types of fibers to be treated. For most of the wool fibers, the common and environmentally friendly bleaching agent is hydrogen peroxide of a 6% concentration. Some yellow wools may require a harsher bleaching, in which cases sodium dithionite or sodium formaldehyde sulfoxylate are used, with a resulting polluting impact made by the effluents.

### 5.4. Stabilizing Dimensions

Wool fabrics are dimensionally stabilized by using at least one of the following setting operations: crabbing and wet and dry decatising. The chemistry behind the process is based on the breaking and reformation of the interchain cystine bonds in the desired places by using a mechanical stretch of the fabric in a hot, wet environment, followed by a cold, wet shock and the release of the mechanical stress. The temperature difference of the shock is usually around 80–100 degrees Celsius. The time required for stretching in the hot, wet environment (water or steam) depends on the weight and density of the fabric, going from 2 to 5 min in saturated steam of 2 bar (120 °C), and from 30 to 50 min in hot water of 98 °C.

An alternative operation is chemical setting, which can cope with a larger amount of cystine bonds than any of the above processes, thus producing long-lasting effects. It makes use of thio-glycols under a stretching operation, followed by hydrogen peroxide reformation of the di-sulfide bonds, like in permanent waving of hair. The process does not require a temperature shock, but the chemical reagents required make it, however, too expensive relative to the usual end-products.

### 5.5. Surface Treatment and Coating and Infiltrating Operations

Surface treatments and coating and infiltrating operations cover the *coloring* and most of the wet-finishing processes. The *coloring* process renders most of the commercial value of a fabric. It is achieved either by coating (*pigment printing*) or by infiltrating (*printing* and *dyeing*) the fibers. While pigment printing binds by the action of a resin coating the coloring pigments, which have no affinity for wool, the other printing systems and the dyeing make use of molecules which diffuse and eventually bind on active sites of the fiber or aggregate inside the fiber to form difficult-to-remove clusters. The dyes used for wool fibers are the same as those from practically any dyeing class; it is only the disperse dyes which are of low interest for wool dyeing, due to their poor hydrophilicity and affinity, and this class remains dedicated exclusively to chemical fibers. The coloring process, no matter which class of dyes is used, pollutes the water with dye molecules that were not successfully bonded by the fiber, as the yield of dyeing cannot reach 100%. A coating operation is usable not only for pigment printing, but also for covering the fibers with certain chemical products which may impart new properties to the end-product, like, for example, an improved easy-care finish, as discussed further on.

The wet-finishing operations comprise *softening, milling* and *easy-care treatment*. The softening process is achieved by using cationic or silicon-based surfactants (softeners) to improve the handleability of the fabric, particularly as to the smooth touch. The milling process makes use of the unique ability of wool to felt, because of the scales on the fiber surface which enable the directional frictional effect. The process is carried out by mechanical action applied to the fabric in a wet environment of controlled pH, with the help of ethoxylated surfactants, and produces the specific wool-like look of the material. The ability of wool to felt has a downside: the machine-washing and tumbler-drying of wool clothes at home change (shrink) the product dramatically. As a result, home easy-care operations are not possible for wool products, unless they receive a specific easy-care finish which can arrest the felting property of wool fibers. The easy-care treatment aims to erode the scales on the fiber's surface and/or arrest the fiber-to-fiber movement using chemical resins. The eroding of the scales is achieved by Allwörden's reaction, which involves a controlled cold chlorination followed by the neutralization of residual chlorine with sodium sulfite [55]. Resin is further applied in order to improve the results. The process is a heavily polluting one, as it releases large amounts of halogenated organic compounds (AOX) in the effluent, as well as gaseous chlorine to the atmosphere, for which reasons new methods are under investigation. Treatments with enzymes, with ozone, or with plasma, followed by resin application, are some of the available alternatives to chlorination. It is of interest to note that researchers identified that the wool fiber's ability to felt is a heritable trait, one dependent on fiber diameter and curvature [56]. Their results suggest that it would be possible to breed for naturally shrink-resistant wool, producing, on the back of the sheep, wool for easy-care products which would not need any special further treatment [57].

*5.6. Treatments for a Finished Aspect*

The treatments for a finished aspect are, generally, mechanical operations which belong to the dry-finishing routine, and comprise raising, shearing, pressing (ironing), and steaming. The dry-finishing operations make use of vegetal or metal-made brushes for raising; helicoidal blades for shearing; hot cylinders, or plates, for pressing; and steamers for steaming. These operations are used to give the final aspect to the fabric's surface, and their main polluting product is the wool powder, formed mainly at shearing and collected by the vacuum cleaners of the finishing machines.

## 6. Pollution in Wool Processing

As previously described, conventional wool processing uses many polluting chemicals, consumes large quantities of water, and generates a large amount of wastewater. In agreement with the sustainable development goals [58], efficient processes need to be applied, with reduced consumption of chemicals and energy, safer solvents, and reduced amounts of by-products [59].

The main polluting steps of wool processing are shown in Scheme 2 [60].

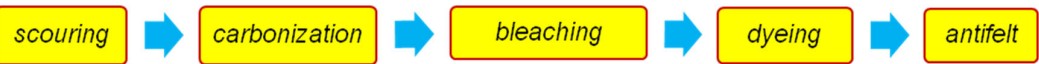

**Scheme 2.** The main polluting steps of wool processing [60].

In some of these steps, the polluting chemicals may be replaced by enzymatic treatments [61]. The replacement of conventional procedures with enzymatic ones leads thus to decreases in the chemicals and energy required for the process and lowers the amount of byproducts, due to the specificity of the enzyme.

## 7. Biotechnology in Wool Processing

Biotechnology replaces polluting chemicals with enzymes and represents an opportunity for future textile developments from the environmental, economic, and even public health points of view [62].

Enzymes are Proteases with catalytic activities. Their classification is based on the type of reaction catalyzed. For the textile industry, the types of enzymes mostly used are Oxidoreductases (class 1) and Hydrolases (class 3) [63]. A number of properties, like high catalytic activity, specificity for reaction, substrate regio- and stereo-selectivity, ability to work in mild conditions, and biodegradability, recommend the use of specific enzymes for sustainable processes. The representative enzymes applied in wool processing are presented below.

### Enzymes for Wool Processing

The most used enzymes in wool manufacture are *Hydrolyses*, enzymes that fragment different substrates by reaction with water. From among these enzymes, representative examples are detailed in the following.

*Proteases* are enzymes which break the chains of proteins. Proteins are biopolymers having as synthons amino-acid molecules. The structure of the protein is complex, being characterized by primary, secondary (α-helix and β-sheet), tertiary, and in some cases, quaternary structures (see Figure 3) [64].

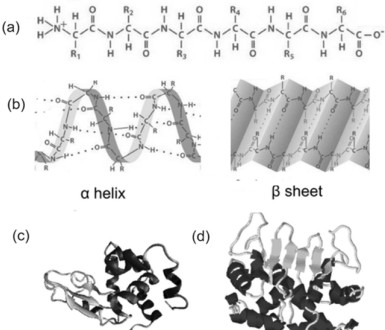

**Figure 3.** Protein structure: (**a**) primary, (**b**) secondary, (**c**) tertiary, and (**d**) quaternary [64].

The *Proteases* may be classified according to their active center structures as *serine-*, *cysteine-*, *threonine-*, *aspartic-*, *glutamic-*, or *metallo-Proteases*, as well as *asparagine-Peptidelyases* [65]. The most widespread enzymes are serine-*Proteases*, which have a triad in the reaction center consisting of the following amino acids: *serine* (Ser), *histidine* (His), and *aspartic acid* (Asp) (see Figure 4). This splits the amide moiety of the protein by transforming it into a carboxylic acid (RCOOH) and an amine (R'NH$_2$).

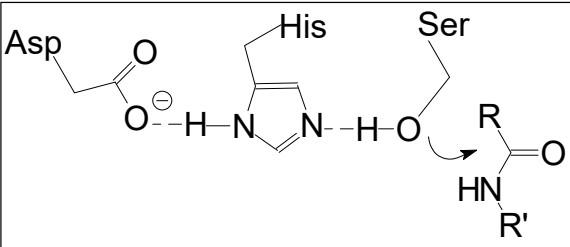

**Figure 4.** Catalytic triad of a serine-Protease.

To this class belong *Trypsin, Chymotrypsin, Elastase*, and *Subtilisin.* The cysteine-*Proteases* act as a dyad comprising a *cysteine* as nucleophile and a *hystidine* as base deprotonating the *cysteine. Papain* and *Bromelain* belong to this class. The threo-

nine-*Proteases* act with only the *threonine*, which has both OH and NH$_2$ groups involved in the catalytic process. *Aspartic-* and *glutamic-Proteases* have acid-based proteolytic mechanisms. *Pepsin* and *Renin* are included in aspartic-*Proteases*, and *Scytalidopepsin B* in the glutamic-*Proteases.* In metallo-*Proteases*, a metal, usually Zn and sometimes Co, is involved in the catalysis. *Matrix Metallo Proteases* are examples of such enzymes.

*Proteases* are involved in the processes of wool finishing, mainly with the antifelting treatments. A specific group of *Proteases* are the *Keratinases*. Keratin, the wool protein, is not easily degraded, due to the disulfide bonds [66].

*Keratinases* hydrolyze the wool protein chain. These enzymes, according to the reaction center, are *serine* or *metallo-Proteases*. For transforming keratin, beside *Keratinases*, an Oxidoreductase, namely, *Disulfidereductase* (EC 1.8.1.8), is needed [67]. In fact, the process of keratin degradation starts by the breaking of the –S-S- bonds, as catalyzed by the redox enzyme, resulting in –SH groups. This transformation changes the stereochemistry of the chain, making possible the access of the *Keratinase*, which may hydrolyze the amino acid chain either internally (*endo*) or at the N/C terminal (*exo*) [68].

*Cutinases* (EC 3.1.1.74) are *Esterases*, reacting with esters and also presenting the catalytic triad *Ser-His-Asp* [69]. These enzymes act on the wool cuticle's hydrolyzing lipids and protein bonds and thus reduce the hydrophobicity of wool [70].

*Cellulases* are *Hydrolyses* which fragment cellulose, a 1–4 β glucose polymer. The bond cleavage is performed by an acid–base catalysis involving two dicarboxylic amino acids working in tandem [71]. Cellulose is found in the vegetal impurities accompanying raw wool. The classical procedure used for eliminating them is by means of treatment with concentrated sulfuric acid, as previously described. Such a process is very polluting and its replacement with an enzymatic treatment is desirable. The enzymatic removal of vegetative impurities on raw wool is known as bio-scouring. There are three types of *Cellulases*: *endo-* and *exo-glucanase*, which hydrolyze the cellulose chain inside or at the ends, and β-*glycosidase*, which hydrolyzes cellobiose, the glucose dimer. These types may be used together or, for more efficiency, mixed with *Pectinases* [72].

*Lipases* of microbial origin (EC 3.1.1.3) are also *Hydrolyses*, ones specific to lipids, catalyzing the hydrolysis of long-chain triacylglycerols. These enzymes may be used for eliminating grease from wool. Microbial *Lipases* belong to the *serine Hydrolyses* group [73].

*Laccases* (EC 1.10.3.2) are *Oxidoreductases* which improve the dyeing process by the polymerization of different phenols on the textile fibers, generating the chromophore compounds. The polymer remains fixed on the material, avoiding the loss of color with time [74]. *Laccases* are *metallo*-enzymes with four copper atoms in the reaction center which transport the electron during the redox process [75]. The huge number of applications of *Laccase* (dyeing and finishing of wool, discoloration of wastewaters) leads to the optimization of preparation procedures for obtaining a mass production of this enzyme [76].

The use of other enzymes, like *Collagenases* [77] or *Transaminases* [78], has also been mentioned.

*Transglutaminases* (EC 2.3.1.13) belong to the *Transferases* group, comprising enzymes moving the acyl group based on *glutamine* to different amines including the diamino acids from wool fibers. The treatment repairs previous damage instances by cross-linking, and improves dye fixation [79]. The discovery of microbial *Transglutaminases* enlarged their industrial applications due to facility in separation (extracellular enzymes) and activity in large intervals of pH (5–8) and temperature (40–70 °C) [80]. Besides the improvement in yarn resistance, the treatment increases wool softness and reduces pilling and felting [81].

The inconveniences associated with the applications of enzymes consist in their lower stability and sensitivity to the reaction conditions, and their relatively high cost. One solution for improving enzymatic treatments is enzyme immobilization [82]. The immobilized enzyme is more stable and can easily be separated and recycled. The pro-

cedures for immobilization are performed by physical or chemical methods (see Table 1) [83].

**Table 1.** Immobilization procedures for enzymes [83].

| Methods | Advantages | Disadvantages |
|---|---|---|
| *Physical* | | |
| *Adsorption* | No modifications of the enzyme structure Simple, Regeneration of carrier, Reduced cost | Desorption; Enzyme subject to microbial attack |
| *Entrapment/Encapsulation* | No modifications of the enzyme structure | Diffusion problems; Leakage |
| *Chemical* | | |
| *Covalent bond* | Higher stability | Possible modification of the enzyme structure, leading to reduced activity; Costly process |
| *Crosslinking* | More stable than free enzyme | Possible modification of the enzyme structure, leading to reduced activity |

The possibility of separating and reusing the immobilized enzymes, which would reduce the process cost, endorses their application in numerous fields. The choice of carrier is of great importance.

The carrier's properties under consideration are the following: high stability, biocompatibility, insolubility, possibility of reuse, and lower cost [73]. A list of carriers compatible with plant *Proteases* is presented by Troncoso and coworkers. Natural and synthetic polymers, as well as inorganic oxides or magnetic particles, are described together with the plant enzymes immobilized on these carriers [84]. Immobilization of an alkaline *Keratinase* on chitosan-based carriers improved the thermal and operational stability of the enzyme [85].

To facilitate immobilized enzyme separation, magnetic carriers may be used. Carboxymethyldextran, combined with ions of iron, was employed for the magnetic nanoparticle, the enzyme being fixed covalently with glutaraldehyde or pentaethylenehexamine [86].

Enzymes are obtained from plants, animals, and microbes, the last source occupying almost 90% of the overall enzyme market [87]. Due to the wide area of industrial applications, the preparation of enzymes has lately been improved, with progress in extraction and purification methods, as well as in formulation, resulting in a larger quantity of enzyme production [88].

## 8. Processing Wool by Sustainable Methods

The replacement of conventional procedures with enzymatic ones leads to decreased chemicals and energy usage for the process and lowers the amounts of by-products due to the specificity of the enzyme.

### 8.1. Bio-Scouring

The conventional scouring process described previously may be performed with enzymes like *Cellulases*, *Pectinases*, *Proteases*, *Cutinases*, *Lipases*, and *Collagenases*, which are good for selectively removing vegetable residue and grease from the raw wool.

A mixture of hydrolytic enzymes (*Bactosol WO*, Clariant) was successfully used to remove the wool grease [89]. Comparison of the chemically and enzymatically treated wool revealed a better result for the procedure when using the enzyme mixture as scouring agent [89].

A thermophilic lipase, produced by the bacterial strain *Bacillus aerius*, was employed successfully to process raw wool. It produced good results for scouring and dyeing wool, both in the same bath, thus reducing energy and water consumption [90].

An immobilized *Lipase* was used to eliminate the wool surface lipids without damaging the interior of the fiber. An enzyme, produced by an extremophilic organism, was employed for this purpose, and the immobilization stabilized the enzyme and made its separation and reuse possible after each treatment. In this way, an enzyme covalently immobilized on a polyethylene imine–sericin hybrid may be used up to five times [91]. The same carrier has been used for immobilization of a *Protease* from *Bacillus safensis* FO-36bMZ836779, with treatment studies showing good results as to improvement in wool shrink-proofing [92].

By using *Proteases,* the scouring process is improved and the cuticular layer modified, increasing the dyeability of wool fibers. Elimination of lipids may be performed with *Keratinases*. The immobilization of the enzyme made it reusable and consequently reduced the cost of the process [85].

The treatment with *Cutinases* followed by *Proteases* is more efficient in wool scouring [93].

The combination of the enzymatic treatment with silver nanoparticles improved the results for *Cellulases* but not for *Lipases* [94].

Along with enzymes as scouring agents, ultrasonic irradiation also reduces the environmental impact of the process without damaging the quality of the wool fibers [95,96]. The development of the ultrasonic scouring procedure creates a cleaner and greener process, making the use of wool fiber more sustainable [97,98].

Summing up, the use of enzymes during wool scouring is a viable alternative to the traditional scouring, providing that the mechanical action is retained. The results for bio-scouring establish that, in addition to wax, vegetable matter and undesirable fibers can also be removed, and at a lower temperature than classical scouring, saving energy.

The water of the washing process may be reused after treatment with cationic polyamide to separate the solid waste, from which lanolin is to be extracted, with an ethanol–methanol mixture [99].

Another procedure for wool cleaning without water is based on the use of supercritical (sc) $CO_2$. The white index is increased, and the Gram-positive and Gram-negative bacteria deactivated by treatment with sc$CO_2$ [100].

Wool scouring should preserve as much as possible the properties of the fibers, such as luster, tensile strength, humidity, hygroscopicity, etc. Enzyme scouring, being a milder process than the classical one, best protects the wool fiber's properties. An after-scouring treatment of wool using *Transglutaminase* improves even more the wool's luster, softness, and tensile strength [101].

### 8.2. Carbonization

The classical way, as described previously, to remove vegetable matters contained in raw wool is by a treatment with sulfuric acid. The procedure is very polluting, and many attempts have been made to replace or reduce the quantity of the sulfuric acid. Treatments with enzymes like *Cellulases*, *Pectinases*, *Hemicellulases*, and *Xylanases* have been investigated in this respect. So far, no single enzymatic treatment for wool carbonization has been suggested [93].

*Cellulases* may be used to make the procedure eco-friendly. There were a few papers published on this topic some years ago showing the cellulolytic treatment as having a reduced efficiency [102,103]. A combination of enzymatic and chemical treatments was proposed by Sedelnik [104]. The advantage was associated with the reduction of the quantity of sulfuric acid, which means a lower degree of pollution. A mixture of *Glucanases* (EC 3.2.1.4), acid *Pectinase* (EC 3.2.1.15), and *Xylanase* (3.2.1.8) proved more efficient, being successfully used for wool treatment without sulfuric acid [105]. No recent publications concerning *biocarbonization* were found.

### 8.3. Bleaching

This process eliminates the last impurities and improves the whiteness of the wool. The most used agent is hydrogen peroxide. The procedure is applied under various conditions, either acidic, or alkaline. By means of this treatment, it is claimed that wool obtains a smoother surface than that of the untreated fibers [50].

The addition of enzymes is recommended in order to reduce the bleaching time. For instance, combining bleaching with a *Protease* treatment improves the resultant wool's properties [106].

The hydrogen peroxide may also be produced in situ by using *Glucose oxidase*, which catalyzes the transformation of glucose to gluconic acid and hydrogen peroxide [107] Promising results have been obtained by using this procedure for wool bleaching [108].

### 8.4. Antifelting Enzyme Treatments

One particular property of wool is its ability to felt, due to the scales on the fiber surface, which lead to interlocking of the fibers and shrinking of the material. Shrink-proofing treatments are required in order to make wool materials machine-washable. As has been described above, the treatments either etch the scales and/or cover the fiber surface with a polymer. Such treatments are very polluting due to the use of chlorine.

Hassan and Carr [109] compare several methods used for anti-felting treatments including oxidative, enzymatic, and plasma procedures. The authors stated that the chlorine-Hercoset procedure is the most effective treatment but underlined the negative environmental impact of it. The enzymatic treatment is less efficient and difficult to control, the *Proteases* having the potential to damage the wool fibers.

A new *Protease*, a fungal enzyme AtP produced by *Aspergillus tubingensis*, proved to be successful in reducing felting and eliminating lice eggshells from wool [110].

A mixture of proteolytic enzymes was produced by *Bacillus* sp. 51, containing serine *Proteases* and *Metallopeptidases* in combination with *Oxidoreductases* capable of breaking the –S-S- bonds. These enzymes reduce the felting of wool top with a very small weight loss and no fiber damage [111].

A combination of enzymes produced by *Bacillus* sp. 51 or *Bacillus patagoniensis* PATO5$^T$, when used with a biosurfactant in the anti-felting treatments, reduced the enzyme diffusion into wool by acting mainly on the fiber's surface [112].

*Proteinase K*, which is produced by a fungus (*Tritirachium album Limber*), and easily hydrolyses keratin, was also efficient when used in in shrink-proofing and anti-pilling treatments of wool [113].

A recycled keratin extract treatment proved to be efficient as an anti-felting solution, improving also the softness, whiteness, and dyeability of the treated wool. [114].

Shrink-proofing treatments have been performed with *Proteases* extracted from a plant (*Cynara cardunculus* L.). The treatment ensures a good shrink resistance for wool yarn and fabric [115].

Treatments with *Lacase* or a *Protease*, followed by coating with poly-carbohydrates (chitosan, wheat, gum arabic, or starch), have been performed. The best results for shrink resistance have been obtained by the combination *Protease*–chitosan [116]. The treated wool has a reduced area shrinkage and unaffected tensile and bending properties.

Another enzyme used in anti-felting treatments is *Savinase* (EC 3.4.21.14), which is produced by *Bacillus lentus*. *Savinase* 16 L combined with an organic phosphine [P[(CH$_2$)$_n$OH]$_3$, n = 1 to 10)] as activator, and employed in a multiple short time padding process which may be applied successfully at an industrial level [117].

To avoid damage to the wool fiber by the *Protease*, *Savinase* was covalently immobilized on a complex carrier based on poly (ethylene glycol) bis (carboxymethyl) ether and L-cysteine: HSCH$_2$ NH (COOH) COCH$_2$[OCH$_2$CH$_2$]$_n$OCONH(COOH)CH$_2$SH. A good anti-felting result was obtained by this treatment [118].

Proteolytic extracellular enzyme produced by a thermophilic bacterium (*Bacillus safensis* FO-36bMZ836779) was extracted and used for a wool treatment, both free and immobilized on an activated agar carrier. The treated wool became machine-washable without significant deterioration of the fibers [92].

The proteolytic enzyme treatment, besides its anti-felting effect, improves the whiteness and dyeability of the wool [119].

Mixed treatments have also been tried, some of them being successful. For example, the *Protease* treatment was combined with different pretreatments like corona discharge or plasma, as well as with other enzymes such as *Cutinases*, *Lipases*, and *Transaminases*, and the results in terms of shrink-proofing and whiteness degree were encouraging [120].

Also, the combination of a commercial *Keratinase* with chitosan improves the dimensional stability of wool, as well as its wettability and dyeability [121].

### 8.5. Reduced Pollution in the Dyeing Operation

According to an evaluation of the footprint of the processing of woolen textiles, the dyeing phase is the most toxic for the environment [60]. Textile dyeing wastewaters represent 75% of all wastewater containing dyes [122]. Thus, the application of sustainable, ecologically friendly solutions seems appropriate, and several recent studies have addressed this topic.

One issue relevant to diminishing the environmental impact of dyeing is the use of natural dyes. The classification of natural dyes based on their sources is given in Figure 5 [123].

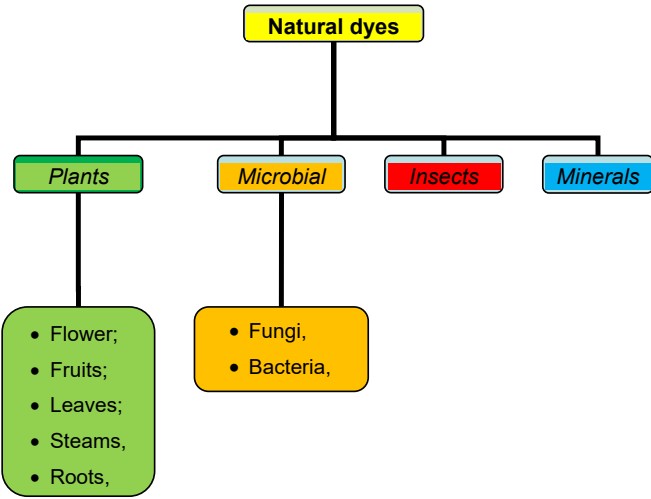

**Figure 5.** Sources for natural dyes [123].

Numerous studies have been dedicated to examining this aspect and, as a result, there are various solutions which have been suggested for dyeing various colors (see Table 2).

**Table 2.** Dyeing wool with natural dyes.

| Source | Color | Procedure | Properties | Lit. |
|---|---|---|---|---|
| Marigold flower as powder | Yellow | Mordants: Alum, Iron or Tin Chloride | Alum and Tin mordant bright color, Iron dark; Light fastness: 5; Wash fastness: 4.5–5 | [124] |
| Onion skin | Dark yellow | Plasma, Alum as mordant, Dyeing by spraying | Wash fastness: 4.0; Reduced energy and water consumption | [122] |

| | | | | |
|---|---|---|---|---|
| Nutshell | Brown | Mordants: Alum, Iron sulfate and Chitosan | Wash fastness: 4–5; UV-protection for skin | [125] |
| Little hogweed | Brown | Mordant: Banana tree leaf | Wash fastness: 4–5; UV and Antibacterial protection; Diminished toxicity (bio-mordant) | [126] |
| North Indian rosewood | Dark red | Mordants: Aloe Vera and Amla | Wash fastness: 4–5; Antibacterial protection; Diminished toxicity (bio-mordant) | [127] |
| Camphor tree leaves | Brown | - | Wash fastness: 4–5; UV and Antibacterial protection | [128] |
| | Pink Pink–Red Red | Mordants: Arjun, Gallnut, Pomegranate, Citric acid, and Chlorophyl | Wash fastness: 4–5; Light fastness 4–5 (with higher quantity of mordant); Diminished toxicity (bio-mordant) | [129] |
| Black rice | Pink (pH 3) Yellow (pH 10) | Mordants: Copper sulfate and Aluminum sulfate | Wash fastness: 4–5; Light fastness: 6; UV-protection to skin | [130] |
| Kesudo herb Cinnamon herb Goldenrod herb | Yellow–Red Red Brown | -- | Wash fastness: 4–5; Light fastness poor; Diminished toxicity (no mordant) Covers Sustainable Development Goals: 3, 13, and 14 | [131] |
| Cinnamon bark | Yellow | Mordants: Henna leaves, Rose petals, Pomegranate peels, and Turmeric rhizome | Wash fastness: 4–5; Light fastness: 4–5; Rub fastness: 4–5; Diminished toxicity (bio-mordant) | [132] |
| Common basilisk | Yellow to Yellow–Brown | Mordants: Binary combinations of Al, Sn, Cr, Fe, Cu, Ni, Co, and Zn salts | Wash fastness: 4–5; Rub fastness: 4–5; Light fastness: 5–8, depending on mordant type and quantity | [133] |
| Cochineal beetle | Red | Chitosan-polypropylene imine dendrimer | Wash fastness: 3–4; Light fastness: 7 | [134] |
| | Red–Orange | Mordants: Henna leaves and Pomegranate peels; Microwave | Wash fastness: 3–4; Light fastness: 3–5; Diminished toxicity (bio-mordant) | [135] |
| | Red–Orange | Plasma; Dendrimer (Polypropylene imine) | Improved dyeability | [136] |
| Madder | Red (Al, Sn, Ni, Co, Zn); Grey (Fe, Cr); Brown (Cu) | Mordants: Single-metal or binary combinations of: Al, Sn, Cr, Fe, Cu, Ni, Co, and Zn salts | Wash fastness: 4 (Cu, Cr, Al+Cu, Fe+Cr, Fe+Zn, Sn+Cr, Co+Ni, Co+Cr); 4.5 (Al+Cr, Cu+Cr); Light fastness: 8–7 | [137] |
| Madder Gardenia (blue and yellow) | Combination of dyes: Red, Violet Yellow, Green, Brown | - | Wash fastness: 5; Uniformity of dyeing; Diminished toxicity (no mordant) | [138] |
| Madder Cochineal beetle | Red | Mordants: Chitosan and a dendrimer (methyl acrylate + ethylene diamine) | Antibacterial and antioxidant properties | [139] |
| Madder Reseda | Red Beige | Mordant: Pomegranate peel; US: dye extraction | Wash fastness: 4–5; Light fastness: 4–5; Diminished toxicity (bio-mordant) | [140] |
| Madder Weld | Red Beige | Mordant: Oak | Wash fastness: 4–5; Rub fastness: 4–5; Light fastness moderate; | [141] |

| | | | Diminished toxicity (bio-mordant) | |
|---|---|---|---|---|
| Purple cluster geranium | Yellow–Red | Mordant: Tin chloride | Wash fastness: 4–5; Light fastness: 4–5 | [142] |
| Roasted peanut skin | Red | Microwave | Wash fastness: 4–5; Light fastness: 5–6; Diminished toxicity (no mordant) | [143] |
| Henna + *Acacia nilotica* pod (*An*) | Dark red | Chitosan; Mordants: Tannins from *An* | Wash fastness: 4–5; Light fastness: 4–5; Antibacterial activity | [144] |
| Walnut dyes | Brown | Mordant: Alum; Nano Ag,TiO$_2$, Al$_2$O$_3$ | Antibacterial activity | [145] |
| | | Mordants: Acacia bark, and Turmeric | Wash fastness: 4; Light fastness: 5; Rub fastness: 4–5; Diminished toxicity (bio-mordant) | [146] |
| Pomegranate Walnut Green | Beige Brown | Mordants: AgNO$_3$, ZnO and Cu$_2$O | Wash fastness: 4–5; Light fastness: 5–7 | [147] |
| Saffron flower | Brown to Green | Mordants: Alum, Cu and Fe sulfates and Tin chloride | Wash fastness: 4–5; Acidic and alkaline perspiration: 4–5 | [148] |
| Sulfonated kraft lignin | Yellow | Hydrogen peroxide | Wash fastness: 4 | [149] |
| Mugwort leaves | Green Brown | Mordants: Alum, Aluminum chloride, and Ferrous sulfate | Wash fastness: 4–5; UV protecting; Antibacterial | [150] |
| *Talaromyces atr.* TRP-NRC | Red Brown | Genetic modification of the pigment source | Wash fastness: 4–5; Acidic and alkaline perspiration: 4–5; Diminished toxicity (no mordant) | [151] |
| Red grape pomace | Red | Mordants: Tannic acid, Mimosa extr.; US | Wash fastness: 5; Rub fastness: 4–5; Light fastness: 4–5; Diminished toxicity (bio-mordant) | [152] |
| Harmal seeds | Light brown | Mordants: Turmeric rhizomes, Henna leaves, Acacia bark, and Pomegranate peels | Wash fastness: 4–5; Acidic and alkaline perspiration: 4–5; Diminished toxicity (bio-mordant) | [153] |
| Tea leaves' tannins | Brown | Mordants: Pomegranate peels, Acacia bark and Turmeric rhizomes | Good fastness; Diminished toxicity (bio-mordant) | [154] |
| Rose geranium waste | Brown Beige | - | Diminished toxicity (no mordant) | [155] |
| Lac insect (Lacaic acid) | Deep purple | Mordants: Turmeric rhizomes and Acacia bark T | Wash fastness: 3–4; Light fastness: 4–5; Diminished toxicity (bio-mordant) | [156] |
| Melanoid from *Lycium barbarum* | Brown | Pomegranate peels as mordant | Wash fastness: 4–5; Antibacterial; Acidic and alkaline perspiration: 3–5; Diminished toxicity (bio-mordant) | [157] |
| Alkanet (*Alkanna tinctoria*) | Red | Mordants: Acacia, Turmeric, and Pomegranate; US | Wash fastness: 4; Light fastness: 4–5; Diminished toxicity (bio-mordant) | [158] |
| *Trachyspermum coptiocum* | Yellow–Red | Mordant: Date seeds | Wash fastness: 4; Light fastness: 5–6; Diminished toxicity (bio-mordant) | [159] |
| Reseda | Yellow | Mordant: Wild olive | Wash fastness: 4–5; Light fastness: 4–5; Diminished toxicity (bio-mordant) | [160] |

| Wild tumeric | Yellow | Mordants: Acacia, Pomegranate, and Pistachio | Wash fastness: 4–5; Light fastness: 5; Diminished toxicity (bio-mordant) | [161] |
|---|---|---|---|---|

For fixation of the natural dyes the dyeing needs to be performed on pre- or post-mordant-treated wool. The mordant is either a metal salt or an organic compound containing OH functional groups. The dye fixation is performed with the metal ion in the metal salt case and with the functional groups in the bio-mordant case. The fixation of the dye is due to the bonds created between the wool surface and the dye (see Figure 6).

**Figure 6.** Dye fixation to wool through a mordant.

Also, ionic bonding with the COOH group from the dye is possible in the cases of dendrimers (Den) used as mordant (see Figure 7).

**Figure 7.** Dye fixation to wool through a dendrimer with amino terminal groups as mordant.

Usually, the natural dyes have oxygen for coordination with the mordant, either as OH or as C=O groups (see Figure 8).

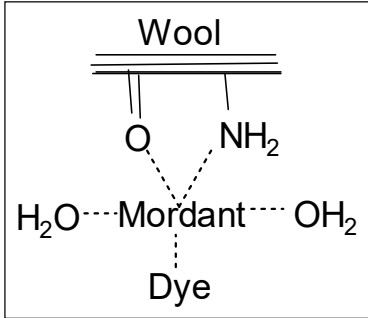

**Figure 8.** Chemical structure of components from natural dyes.

The color hue of natural dye is shifted after complexing with mordants, and different mordants result in different color changes, causing problems in controlling the color stability. Less toxicity and better fixation have been realized by using mixtures of plant extracts [138]. The use of organic mordants is considered ecologically sound and has solved the problem of dye fixation and that of the fastness of the dyed wool.

Another aspect of the making of sustainable dyeing processes is the reduction of dyeing energy. Experiments have been performed studying dyeing with natural dye by using green energy, and a reduction of 24% of the power supply was realized [162].

Reduction of dye pollution was also achieved by the synthesis of the dye on the wool fiber through polymerization by oxidation of small organic compounds. Those include a dyeing process in the presence of a *Peroxidase* (EC 1.11.1.7), using various organic compounds like 1,4-dihydroxybenzene, 2,7-dihydroxynaphthalene, catechin hydrate, 2,5-diaminobenzenesulfonic acid, 3-amino-4-hydroxybenzenesulfonic acid, 1,4-diaminobenzene, 2- and 4-aminophenol. The dyeing proved to be efficient under mild conditions without damage to fibers, and with good fastness to washing and light. Moreover, depending on the pH used, different colors and shades may be obtained [163].

One of the catalysts often used for dyeing by means of the polymerization of small molecules is *Laccase*. Some examples are presented further on.

Wool was dyed by *Laccase* polymerization of aromatic phenols or amines. Brown shades were obtained with 1,4-dihydroxybenzene or 2,5-diaminobenzenesulfonic acid, and beige-to-green with 2,7-dihydroxynaphthalene. Very good wash fastness was obtained for all the dyed materials [164]. 1,3-Dihydroxybenzene treated with *Laccase* together with 2,2′-azino-bis(3-ethylbenzothiazoline-6-sulphonic acid) dyed wool, giving a material with good fastness properties. The dyeing bath may be reused (six cycles); the cost of the dyeing process is thus reduced [165]. Starting from pyrrole, by treatments with *Laccase*, wool was dyed. The resulting polypyrrole was fixed to the wool fiber by covalent and hydrogen bonds. The color depends on pyrrole quantity, going from green to black. The dyed wool has good wash and light fastness and is conductive, and therefore suitable for technical textile applications [166]. A pale-yellow color was obtained by treatment of wool with ferulic acid in the presence of *Laccase*. The dyed wool has a wash fastness of 4–5 and improved antioxidant and deodorizing activities. The light fastness showed poor values (3), the dyed wool being not fit for applications demanding high light-fastness values [167]. Dyeing wool by 2,5-diaminobenzenesulfonic acid polymerization catalyzed by *Laccase* was the basis of a series of experiments which used different pH values. The color obtained depends on the pH value, ranging from dark purple (pH 1.8) to yellowish-brown (pH 10). The structure of the polymer is complex (linear polymer of 2,5-diaminobenzenesulfonic acid including phenazine moieties) [168]. Gallic acid and catechin are also used for wool dyeing by *Laccase* catalyzed reactions. The resulting polymer is fixed in the wool fiber by covalent bonds. Better properties for the dyed wool are obtained when polyethyleneimine (PEI) is added in the dyeing bath. It seems that PEI, by binding through hydrogen bonds to wool fibers, confers a shrink resistance [169]. Caffeic acid polymerized in the presence of polyethyleneglycol gave a brown shade to wool. The material has better wettability and electroactivity [170]. Syringic acid was used as monomer, in the presence of *Laccase* and dyed wool pretreated with 1-ethyl-3-(3-dimethyl aminopropyl) carbodiimide hydrochloride. A deeper yellowish-brown color and better fixation of the polymer on the wool resulted [171].

The *Oxidoreductases* are also efficient in improving dyeing with natural dyes. One of the problems regarding natural dyes is the relatively reduced washing fastness compared with synthetic dyes. Bai and coworkers [172] have applied *Laccase* catalytic polymerization for dyeing wool to gallnut, grape seed, and turmeric extracts, obtaining a material with good washing fastness. Green tea extract with *Laccase* is the basis of an efficient wool dyeing procedure by the polymerization of phenolic compounds from the extract. The dyed wool has antibacterial, antioxidant, and UV-protective properties [173].

Enzymes may be used for improving dye fixation by modifying the fiber's surface. Madder as dye with bentonite as mordant has been applied to wool pretreated with microbial-*Transglutaminase*. The enzyme catalyzes cross-linking lysine-glutamine on the wool fiber's surface, helping the fixation of the dye. An increase in the washing fastness properties was observed for the wool pretreated with enzymes [174].

Another enzymatic pretreatment of wool fiber was applied in dyeing with bloodroot seeds (beige) and madder roots (red) extracts. The pretreatment enzyme was a mixture of *Amylases*, *Lipases*, and *Proteases* produced by animal pancreas. The madder root extract contains salicylic, ellagic, and benzoic acids and quercetin, with OH, C=O and COOH functional groups that are expected to attach on wool [175].

Other enzymes used for the modification of the surface of wool fabric are keration-lytic *Proteases* produced by *Streptomyces harbinensis* and *Streptomyces carpaticus*. This treatment, coupled with functionalization, helps to provide UV protection, antibacterial activity and washing fastness [176].

Enzymatic pretreatment also improves dyeing with synthetic dyes [177]. Protease and sodium alginate treatment improved wool dyeability with reactive dyes [178].

Enzymatic treatment may help the surface patterning in wool/polyester blends, patterns in relief resulting due to wool degradation by the *Proteases* [179].

Besides the choices of dyes and pretreatments, water consumption represents a problem in the dyeing process. Recent publications analyzed this aspect and proposed some ecologically sound solutions. For instance, the water from bleaching operation may be further used for dyeing, with consequent improvements in the dyeability [180]. The reduction of water for the dyeing has to be accompanied by the addition of auxiliaries in order to prevent dye aggregation, which would lead to an uneven dyeing [181].

Some studies have revealed that wastewaters may be successfully reused, thus reducing water consumption [182].

Magnetically treated water presents another opportunity for dyeing wool, reducing the dye consumption, and improving dyeing quality [183].

Engineered water nanostructures were efficient in improving the pilling-resistance of knitted wool fabric [184].

### 8.6. Bio-Treatments for a Finished Aspect

New properties of a wool may be developed by grafting the fibers with different compounds. Enzymes play an important role in this process. Enzymatic treatment with *Laccase* to change the wool yarn shape by the grafting of tyrosine was successfully performed [185]. Wool grafted with β-cyclodextrin was prepared using *Laccase* as catalyst [186].

Ecologically sound solutions have also been considered for imparting anti-bug properties to textiles, and natural, biodegradable compounds are used for this finishing operation [187]. For example, the mothproofing treatment of wool with nano kaolinite showed promising results [188], and rare earth ions addition proved efficient in an antibacterial treatment [189].

## 9. Sustainable Aspects in the Valorization of Wool

### 9.1. Textile Products

Wool is primarily a valued textile fiber. The fact that wool cloth is good to wear in all seasons is an important feature which promotes the use of wool for cloth manufacture based on woven or knitted fabrics. An analysis of knitted wool has revealed the variation of the pores (opening and closing) by sweating and shivering to keep the skin dry and give comfort and safety, regardless of the season [190].

Depending on the fiber diameter and length, wool can be used for both interior textiles (medium wool, 24.6…32.5 micron, and coarse wool, with diameters of more than 32.5 microns) and for apparel (fine wool, up to 24.5 microns in diameter, and medium wool), as shown in Figure 9.

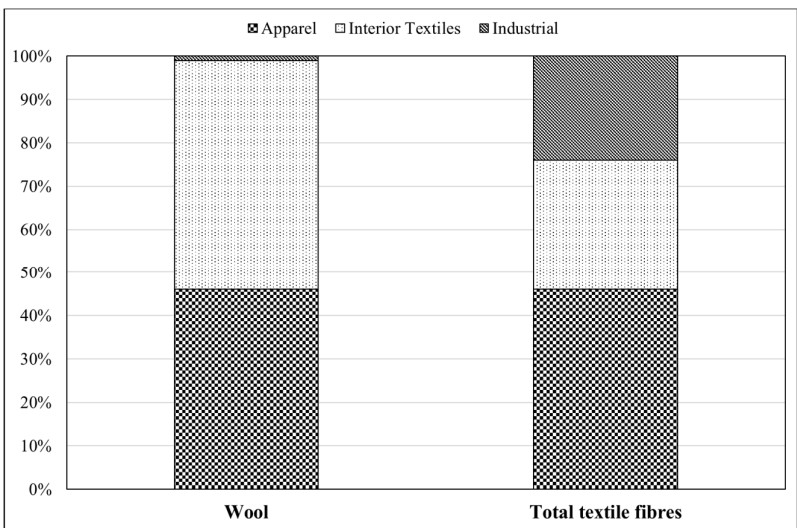

**Figure 9.** World fiber consumption by end-use, in 2022 [11].

The price of raw wool fiber is well above those of the other textile fibers (cotton and other cellulosic and synthetic fibers), being 4 to 10 times higher than any of them [11]. This supports wool's position as a premium fiber for textiles (apparel and interior products) and explains the small amount used for commercially cheaper industrial applications (around 1% of total wool fibers), like felts for filtering, or house insulation, which is described further on.

A strong type of wool is needed for carpet production, which comprises 45% of the world's wool production, New Zealand being the largest supplier [191]. According to Broda and coworkers [192], the coarse wool carpets produced by the tufted technique seem to be very good sound absorbents. The works of Kobiela-Mendrek and coworkers support the same assumption [193].

Knitting is another way of processing wool. The improvement of knitting machines, digitalization, and robotization ensure a sustainable development implementation in this wool-exploitation area [194].

Blends with synthetic fibers have better properties and longer lives. Wool and propylene yarns have a reduced pilling tendency and are thus suitable for winter socks [195].

As discussed above, the processing of raw wool for obtaining woven or knitted materials involves a number of polluting steps. A group of researchers from Norway proposed the reduction of pollution by local production development, which eliminates the pollution of transport operations and develops the transparency of the value chain and the concern for the environment [196].

While wool is primarily used for textiles, for which the diameter of the fiber dictates the end-use, more recent research has put into evidence new fields of interest for which the diameter of the fiber does not play any role. The environmental cost of wool processing has imposed the discovery of new applications in order to justify the expenditure. Thus, a performed *Strengths, Weaknesses, Opportunities, Threats* (SWOT) analysis revealed that wool may be used successfully, instead of plastic materials, in many products other than clothes [197].

*9.2. Other Usages of Wool*

The sustainable development principle has stimulated the occurrence of many technically valuable applications for wool. Some examples of these new wool-applications are presented.

### 9.2.1. Smart Textiles

One such application is the production of *smart textiles* serving as wearable electronic sensors (e-sensors). Thus, wool socks have been produced which can monitor the number of footsteps [198]. Knitted wool coated with silver nanowire may be used as electronic textiles, which can used as e-sensors for body motions [199]. Efficient motion e-sensors have been prepared by coating knitted wool with a suspension of graphene nanoplates, in acetone, by sonication. The prepared sensors are highly stretchable and flexible, being able to build up and to discharge electric charges, property applicable for their functionality as super-capacitors [200].

Studies considering the deposition of conductive components on wool single-jersey have been performed by Wilson and coworkers [201]. The fabric was treated with grapheme ink, reduced grapheme oxide, and three encapsulants. A good interaction between graphene and wool was evidenced, with deposits found on the wool scales, while the encapsulants filled the interstitial spaces. The authors underlined the fact that the deposition of conductive components on the textile depends on the structure of the textile material, no general procedure being applicable.

The successful involvement of wool as raw material for obtaining *smart textiles* for super-capacitors is mentioned by Grube and coworkers [202]. Merino wool, used with nylon and coated with conductive polymers like poly(3,4-ethylenedioxythiophene) and poly(styrenesulfonate), enables the fabrication of good-quality *smart textiles* for sensors [203].

Highly conductive wool fabrics have been prepared by a multi-step process by coating wool with an ink based on graphene nano-platelets and particles of carbon black. Low voltage heating devices have been fabricated from such products [204].

A substrate based on wool with silver nanoflower deposition was employed for a successful surface-enhanced Raman scattering measurement of traces of the pesticide Carbendazim [205].

### 9.2.2. Agriculture Applications

Wool is a perfect source of carbon and nitrogen. Thus, sheep's wool may be used as substrate for the cultivation of different plants. For example, the use of sheep's wool in cucumber cultivation reduces the water consumption as well as the contributions to the greenhouse effect [206].

Pellets from sheep's wool are efficient bio-fertilizers which can be used for organic vegetable production. Experiments performed with tomato and spinach cultivation showed wool pellets and commercial fertilizers to result in similar production levels [207].

Coarse wool was applied as fertilizer to a bean cultivar (*Phaseolus vulgaris* L.), decreasing the *carbon footprint* by 10% [208].

Mulch mats obtained from wool prevent soil erosion, gradually release moisture into soil, and promote the growth of planted vegetation, stopping the growth of weeds [209]. Wool was productively used as vegetation mats for eight perennial plants. The content levels of carbon and nitrogen, as well as the water storage capacity, recommend wool as a vegetation mat for landscapes [210].

Wool is also a source of plant biostimulants, as a substrate for the cultivation of keratinolytic fungi such as *Paecilomyces lilacinus* 112 [211].

### 9.2.3. Building Sector

One sector requiring carbon policy actions is the building sector. Due to climate change, the decarbonization of this sector is compulsory. New biodegradable materials have to be considered, given their sustainability. Wool is such a material. Its properties like breathability, moisture management, and low heat and sound conduction, together with natural flame-retardancy [9], make wool a useful material for house insulation. For

the efficient application of wool as a building insulator, a study identifying the best locations for wool collection centers is of great help, reducing transport pollution and achieving the sustainable development of the region [212].

Besides its thermal properties, wool is a good acoustic insulator. Non-woven wool products may be used in multi-layer room insulation, its sound adsorption depending on surface density, thickness, and air permeability [213]. Valorization of coarse wool fibers for insulation in the context of eco-building was presented by a team of Romanian researchers. Many benefits may be obtained by good local management [214]. Good insulators were obtained by combining wool not appropriate for textile manufacture with hemp. Compared with other insulating panels, wool–hemp panels are a sustainable solution capable of decreasing the environmental impact [215,216]. The combination of wool with agricultural waste, like sugar bagasse, provided excellent insulators. Natural fibers are a good option for replacing synthetic materials in buildings [217]. Insulating properties have been improved by doping the fibers with boron. The new material is a good thermal and sound insulator for buildings [218]. Sheep's wool soy protein bio-composite is another efficient acoustic insulator [219].

Solid bricks presenting good thermal insulation and mechanical resistance have been obtained from wool fibers and clay. Wool fibers have an important effect on this new biomaterial, leading to higher energy efficiency [220]. Wool composite with natural rubber or polyurethane acrylate-based resin makes another good thermal and acoustic insulator material [221]. Complex mixtures containing polyester, date-palm fibers, and wool have been prepared. The new material is a good-performing and low-cost insulator that may be applied in building systems, home furniture, automotive parts, etc. [222].

Other insulators were obtained from wool and cement mortar. The resulting panels have good thermal and mechanical properties [223]. The incorporation of wool fibers in magnesium phosphate cement leads to the optimization of thermal comfort and energy efficiency. The material may be applied as an external thermal insulator [224].

According to the literature, with the use of bio-based insulators in France, by 2050, a saving of 75,000 tons of fossil fuels and a reduction of the greenhouse effect to the order of 312,771 tCO2eq is expected [225].

### 9.2.4. Materials for Industrial Applications

Wool may be used as a constituent of new biomaterials comprising a large field of applications. This is justified by the number of properties of the wool fibers recommending them as components in composite materials, like durability, ability to flex, fire resistance, etc. [226]. Blends with other materials may have various applications. A combination with palm fibers, as nonwoven material, was used for automotive interiors [227]. A mixture of soy protein and wool was used in the manufacture of membranes for lithium batteries [228].

Wool fibers may be included successfully in railway brake shoe composition, imparting a low cost, and lowering the environmental impact [229].

Wool blends show good results in coating anode electrodes. According to the experimental results, such an anode is suitable for photo-catalytic fuel cells, as well as for wearable electronics [230].

A combination of the expensive clay used as subgrade material in roads with wool and banana fibers provides a new, less costly composite which maintains a high level of the road's carrying capacity [231].

Bio-composites obtained from wool fibers and wheat gluten present good mechanical and fire-retardancy performance, and are useful as fire protection materials [232].

Good results have been obtained by combining wool fibers with pineapple leaf or sisal fibers in a natural rubber matrix. The new materials may be applied for packing and other household purposes [233].

Wool's ability to retain various chemicals has been exploited for centuries, the legend of "the golden fleece" being based on the real fact of the utilization of wool in screening

gold particles from flowing rivers. This property is now used for making wool filters for purifying waters used in the food or pharmaceutical industry, wool filters being able to absorb even the traces of heavy metals present in waters [234,235]. Raw wool is also a good metal adsorbent for metal ions like Zn and Cu, with a sorption efficiency of over 90% [236]. Wool filters reduce the virus content in water better than polymers like polypropylene and polyester or river sand [237]. Wool composite made with magnetite and polysiloxane is a good adsorbent for oil spill removal. By centrifugation, the adsorbed oil is recovered [238]. Wool has also been proved to absorb pollutants from the surrounding atmosphere. In this respect, of particular interest is wool's capacity to absorb formaldehyde [239], tobacco smoke, and various other unpleasant odors [240].

The protein chains of wool fiber sequester cations. This property was mentioned previously as being used for filtering water, but it is also the basis for dyeing wool with chrome dyes. Recently it has been shown that the sequestering of metals by amino acids of protein chains make an enzyme-like structure and may be further used for enabling wool fibers to act as catalysts, or catalyst supports in various complex reactions [241]. This field is newly under development and, with the increased demand for new biocatalysts, one may expect an increased demand for wool as well. Wool, combined with poly(2-amino thiophenol), was used as carrier for palladium nanoparticles. The resulting product may be used as an industrial catalyst [242]. A Fe-Co nanocatalyst was prepared by pyrolysis of the mixture of wool, dicyandiamide, and the corresponding metal salts [243].

### 9.2.5. Keratin Extraction and Applications

Wool is a source of protein, its main component being $\alpha$-keratin. Like all the keratins (e.g., hairs, nails, and feathers) wool fiber is practically insoluble in water under normal conditions, due to its highly cross-linked structure based on disulfide bonds. Temperature and certain chemicals may, however, break these bonds and render wool soluble, producing solutions of polypeptides [244,245].

Several procedures for the extraction of keratin from wool using environmentally friendly processes have been described [246,247]. Preparation methods for obtaining keratin have also been presented by Wang and Tong [99]. The comparative evaluation of five methods for keratin extraction described ionic liquid extraction as having the highest yield (95%) [248]. The necessity to continue the research for new eco-friendly and cost-effective methods for obtaining keratin films from wool and other sources has been underlined in these papers. $\alpha$-Keratin extract has various fields of application (see Figure 10) [249,250].

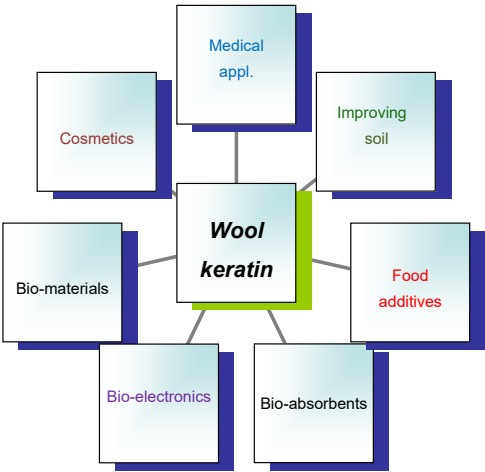

**Figure 10.** Applications of wool keratin extracts.

- *Medical applications*

Because keratin contains several cell-binding motifs of amino acids, it may help cell proliferation, which recommends such films for medical applications, particularly for wound dressing [251–253]. The composites of wool, collagen, and ionic liquids have produced films for wound dressing, drug delivery, and sensors [254]. A very good composite for wound dressing was produced by combining wool with a zirconium-based metal-organic frameworks. The material obtained has air permeability, wash-robustness and moderate hydrophilicity [255].

Keratin based materials may also be applied in tissue engineering. The properties of keratin, like degradation rate, mechanical properties, porosity, swelling, and wettability, support this biomedical application [256].

The material derived by the electrospinning of polycaprolactone with keratin appears to be good for bone engineering [257]. Keratin mixed with bioactive glass and polycaprolactone provides a material suitable for bone repair [258].

Keratin nanofibers obtained by electrospinning are good components in polymer blends for bio-medical applications [259].

Keratin has been developed into hydrogels used as bone scaffolds, wound dressing, and wearable medical devices [260].

- *Biomaterials*

Bioplastics have become of great interest in the context of green sustainable solutions. Keratin is a suitable partner for obtaining such materials, and keratin extracted from wool is a source for such new biomaterials [250]. The advantage is in the biodegradability of these materials, keratin being easily biodegradable. Keratin films, under composting conditions, degrade in 5 days [261]. The combination of keratin with various compounds leads to new improved materials. By grafting keratin with lipoic acid (see Figure 11a) different films may be produced. Lipoic acid binds to keratin by amide bonds, conferring a new geometry on the material obtained. This grafted keratin is more thermally stable and has improved physical properties [262]. Combination of keratin with polybutylene succinate (PBS) (see Figure 11b) resulted in nanofibrous materials successfully used for drug delivery and scaffolds for cell development [263].

**Figure 11.** Compounds for keratin grafting: (**a**) lipoic acid and (**b**) polybutylene succinate.

- *Bioelectronics*

Wool keratin is employed in bioelectronics in such applications as quantum dots for the fluorescent detection of toxic metal traces (ions of chromium and iron). The multiple functional groups (CONH, COOH, NH$_2$) of the keratin may interact with the metal by coordination or ionic bonds [264]. Keratin extracted from wool (WK) may bind on carbon nanotubes (CNT), giving dispersed inks used in high performance bioelectronics (flexible circuits or health monitoring electrodes) (see Figure 12) [265].

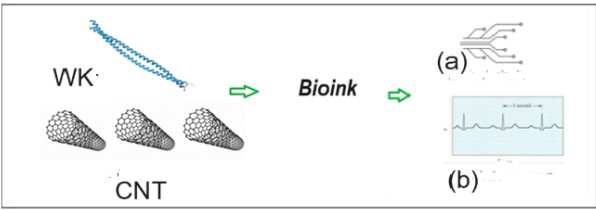

**Figure 12.** Quantum dots based on wool keratin (WK) with (**a**) electronic circuits; (**b**) electrocardiogram [265].

New fibers are obtained by combining wool keratin with a polyanion (poly 4-styrenesulfonate), followed by centrifugation and dry-spinning of the mixture. Due to the outstanding mechanical properties, ion conductivity, and humidity awareness these fibers may be successfully used in strain sensors [266].

- *Food additives*

Wool keratin may also be used as a substituent for casein in food manufacture [267]. Wool keratin food supplements may improve performance in the growth parameters of animals. Such results make keratin a sustainable protein source [268].

- *Biosorbent*

The membrane obtained by combining keratin and polyamide 6 may be doped with silver, Ag, resulting in a good absorbent which can be used in air purification [269].

- *Soil improvement*

Good results in soil fertilization are obtained by mixing biochar obtained by the pyrolysis of lignocellulosic residues (carbon source) and hydrolyzed wool (nitrogen source). This combination gives a C/N ratio corresponding to those of efficient fertilizers [270].

- *Cosmetics*

Keratin is suitable for skin application due to hydrophilicity and the capacity to form films. A film of keratin applied on the skin offers a silky sensation [271]. Keratin particles are proficient as components of cosmetics for hair care [272]. Thus, keratin is successfully used in lotions, shampoos, and conditioners, and for skincare products [273,274].

Concerning the sustainable aspects of wool valorization and the prospects of wool for replacing plastic materials, we have taken into consideration the advantages and disadvantages of such replacement (see Table 3) [197].

**Table 3.** The advantages and disadvantages of using wool instead of plastic materials.

| Advantages | Disadvantages |
|---|---|
| Biodegradability | Produced in low quantities |
| Good acoustic and thermal insulator | Processing for textile materials still polluting |
| Hygroscopic capacity | Classification of coarse wool as a by-product by the EU resulted in less restrictions on disposal (mainly landfill) |
| Odor-preventing capacity | More expensive than synthetics |
| Ensures comfortable sleep | Unsuitable management for collection and processing |
| Hypoallergenic | Difficulty in standardizing production |
| High content of carbon and nitrogen—good fertilizer | EU policy directed more to bioplastic than to natural resources like wool |
| Nontoxic | |
| Low flammability | |
| Potential pesticide (snake, slug) | |

As listed in Table 3, wool has many qualities useful for successfully challenging the use of plastics, and, for the environment, it would be beneficial if wool at least partially replaced these materials.

The sustainability of wool valorization is even more underlined by the many links between the non-polluting processing of wool (e.g., enzymes and natural dyes) and the United Nations Sustainable Development Goals, SDG, as listed in Table 4.

**Table 4.** Sustainable Development Goals (SDG) and the application of environmentally safe procedures for wool collection, processing, and disposal.

| Procedure to Be Improved | Related SDGs |
| --- | --- |
| Sheep breeding and genetic selection | 1, 2, 8, 15 |
| Management of wool collection and distribution | 9, 11 |
| Application of new "green" technology for wool processing | 3, 6, 7, 13, 14 |
| Regulations and economic policy | 15, 17 |

## 10. Conclusions

Pollution generated by synthetic polymers has increased the interest in natural fibers like wool which, due to its properties, has seen growth in consumer preferences. This review surveys the state of the art in wool production, processing, and applications in the context of using less-polluting technologies that make use of enzymes.

There are several issues which need to be underlined when discussing the development of the environmentally friendly wool chain, namely:

- The necessity to develop sustainable sheep breeding for increasing profitability and obtaining an appropriate LCA of wool, starting with its production;
- New solutions for sustainable wool processing using enzymes and natural dyes are emerging;
- New non-traditional usages of wool in replacing non-biodegradable materials have been proposed and developed, opening new horizons for wool of any fiber, regardless of its diameter;
- It has to be stressed that, for any further development, the preservation of the environment has to play a role more important than the purely economic aspects;
- In maintaining sheep breeding and the processing of wool and its use, the social aspects have to be also pondered and included in the whole wool chain;
- An EU policy for encouraging the collection, processing and choice of suitable applications for coarse, processed, and waste wool.

**Author Contributions:** There were equal contributions made by both C.P. and M.D.S. as to conceptualization, data curation, and the writing of the original draft. All authors have read and agreed to the published version of the manuscript.

**Funding:** This research received no external funding.

**Conflicts of Interest:** The authors declare that the research was conducted in the absence of any commercial or financial relationships that could be construed as a potential conflict of interest.

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
