# Peer review of "Eco-Friendly Processing of Wool and Sustainable Valorization of This Natural Bioresource"

_sustainability, doi:10.3390/su16114661_

Round 1

Reviewer 1 Report

Comments and Suggestions for Authors

The use of wool to replace polluting synthetic fibers is a significant challenge, on which we have also contributed from our research team. Since it is a review and the topic is complex, I suggest adding a table of contents. Additionally, it seems to me that the hypothesis on which the work was done is not clear; it should contain specific answers to: Is it feasible to produce wool bioproducts that can replace synthetics? What does wool lack to be competitive as a textile and non-textile raw material to advantageously replace synthetic equivalents?

Some suggested points where the reviewed aspects can be improved: page 5; lines 145 to 146: The behavior of goats while grazing is not clear: firstly, they are not always in the same places as sheep, but rather browse bushes; secondly, it produces two animal textile fibers that can be used for the same purpose as this study proposes; unfair competition would not be good advice. Page 5, lines 196-199: Felting is a promising aspect of textile processing to be improved in research; it would be interesting to conduct research with the device called a "felting needle," which is largely unexplored in the industry and mainly used in crafts. The same in page 6: 196 - 199.

Page 11, lines 365-367: The use of enzymes as replacements for synthetic detergents in wool scouring has not yet yielded promising results, but combining them with physical processes such as scouring may be viable. In addition to wax, it removes vegetable matter and undesirable fibers. The same for page 12: 403 - 407.

Page 13: 464 - 466: In the use of natural dyes or colorations, the use of the pigments inherent in pigmented wools has been overlooked. While a significant portion of sheep are non-pigmented (white), the number of pigmented ones could be increased through genetic selection.

Author Response

view Attached file

Reviewer 2 Report

Comments and Suggestions for Authors

Authors have talked about wool product fabrication and its sustainable valorization. It is good research as the authors take into consideration the full outlook on sustainability and not just promote a bioresource. 

1. The very first sentence talks about microplastics being a menace, true but need to backed by literature. Example of review you can refer and cite would be: https://doi.org/10.3390/su152215758. 

2. Is there any literature associated with Life Cycle Analysis of such wool products? if so, please cite. an example would be: https://doi.org/10.1016/j.jclepro.2023.137877

3. Topic wise details and discussion is good but to elevate this manuscript from a report into a review article, please consider adding another section to the last: Discussion and future implications/possibilities. This way, the review can critically analyze the entire process and aspects discussed in this paper. 

Author Response

Reviewer 2

Authors have talked about wool product fabrication and its sustainable valorization. It is good research as the authors take into consideration the full outlook on sustainability and not just promote a bioresource.

Q1. The very first sentence talks about microplastics being a menace, true but need to backed by literature. Example of review you can refer and cite would be: https://doi.org/10.3390/su152215758.

A1: Thank you for this suggestion. We included it in the text as well as other references.

Q2. Is there any literature associated with Life Cycle Analysis of such wool products? if so, please cite. an example would be: https://doi.org/10.1016/j.jclepro.2023.137877

A2: Thank you for the suggestion. The mentioned reference was already included in the text.

Q3. Topic wise details and discussion is good but to elevate this manuscript from a report into a review article, please consider adding another section to the last: Discussion and future implications/possibilities. This way, the review can critically analyze the entire process and aspects discussed in this paper.

A3: Thank you again for this useful suggestion. Following it we added two last Tables to sum up this review: Table 3 to list the advantages and disadvantages of wool versus plastics, and Table 4 showing the SDGs connected to the application of wool instead of the usual plastic materials.

Reviewer 3 Report

Comments and Suggestions for Authors

After a thorough study of the manuscript, I conclude that the content of the manuscript does not correspond to the title. The vast majority of the content has the character of an educational text intended for the reader who is not yet familiar with the issue. Many facts are drawn from old sources (1916 - 1931 - 1951 - 1972). New information, which would be expected according to the title of the manuscript, is presented by the authors only at the end, and that too in a very abbreviated form. I do not find enough motives for new green processes.

I recommend not accepting the manuscript in this form.

Author Response

Reviewer 3

After a thorough study of the manuscript, I conclude that the content of the manuscript does not correspond to the title. The vast majority of the content has the character of an educational text intended for the reader who is not yet familiar with the issue. Many facts are drawn from old sources (1916 - 1931 - 1951 - 1972). New information, which would be expected according to the title of the manuscript, is presented by the authors only at the end, and that too in a very abbreviated form. I do not find enough motives for new green processes.

I recommend not accepting the manuscript in this form.

A: Thank you for the comment, but we cannot agree with the Reviewer’s points.

  • This paper is intended to be a review, not a communication of new results. As such, it contains the review of the basic facts (about 5 pages) and the review of the new developments (some 20 pages).
  • The new developments are inherently greener than the classical processes. The old sources mentioned by the Reviewer are milestones of the wool science, and, so far, cannot be overlooked, or replaced with any new reference of the field.
  • Concerning novelty, most of the references (255 out of 270) is from 2010 on, with about 200 being from the last 5 years. We consider that this clearly points out to the new developments in wool research.
  • Concerning the English the paper was read and corrected by a native English person.

Round 2

Reviewer 3 Report

Comments and Suggestions for Authors

The main revision of the v2 manuscript consisted of adding a structured table of contents to the beginning of the manuscript and renumbering the references after increasing the number of sources by 5. In ten pages the manuscript describes the technologies used (parts 2-6) and when reading the text it is difficult to distinguish which information represents the “new green" procedures. The chosen range of wool applications is wide. On the one hand, this corresponds to reality, on the other hand, it leads to a superficial list of items in an almost password-like style. Although it is stated in the Abstract that "This review makes a critical analysis of recent literature data", I do not find anything like that in the manuscript. Therefore, I must state that my opinion about the quality of the manuscript has not changed. The over-ambitious title of the manuscript should be modified, or the information on the classical technological processes of wool processing should be omitted and the notified critical analysis of new processes should be added.

Author Response

We have changed the title of our paper and the final of the Abstract.

We do hope now you agree with the paper.